# Developing Biodiversity Baselines to Develop and Implement Future Conservation Targets

**DOI:** 10.3390/plants12122291

**Published:** 2023-06-12

**Authors:** Alice C. Hughes

**Affiliations:** School of Biological Sciences, University of Hong Kong, Hong Kong, China; achughes@hku.hk

**Keywords:** indicators, convention of biodiversity, global targets, biodiversity framework, data

## Abstract

With the recent launch of the Kunming-Montreal global biodiversity framework (GBF), and the associated monitoring framework, understanding the framework and data needed to support it is crucial. Unfortunately, whilst the monitoring framework was meant to provide key data to monitor progress towards goals and targets, most indicators are too unclear for detection or marking progress. The most common datasets for this task, such as the IUCN redlist of species, have major spatial inaccuracies, and lack the temporal resolution to track progress, whilst point-based datasets lack data from many regions, in addition to species coverage. Utilising existing data will require the careful use of existing data, such as the use of inventories and projecting richness patterns, or filling data gaps before developing species-level models and assessments. As high-resolution data fall outside the scope of explicit indicators within the monitoring framework, using essential biodiversity variables within GEOBON (which are noted in the prelude of the monitoring framework) as a vehicle for data aggregation provides a mechanism for collating the necessary high-resolution data. Ultimately developing effective targets for conservation will require better species data, for which National Biodiversity Strategic Action Plans (NBSAPs) and novel mechanisms for data mobilisation will be necessary. Furthermore, capitalising on climate targets and climate biodiversity synergies within the GBF provides an additional means for developing meaningful targets, trying to develop urgently needed data to monitor biodiversity trends, prioritising meaningful tasks, and tracking our progress towards biodiversity targets.

## 1. Introduction

We are in the midst of a global biodiversity crisis, with estimates of average population decline of 69% across species globally [1]. Drivers of biodiversity decline include the loss and conversion of habitats, the unsustainable use of wild species, and other forms of unsustainable governance [2,3]. However, despite this magnitude of continuing biodiversity loss, these analyses of trends use aggregations of inconsistent data that are not truly representative across regions or taxa [4,5]. The need for better data has been highlighted by the monitoring framework within the global biodiversity framework (GBF), where a series of headline indicators, as well as component and complementary indicators, is used to monitor progress towards the targets of the GBF. However, this monitoring framework and the NBSAPs (National Biodiversity Strategic Action Plans) require consistent data, and understanding the shortcomings, gaps, assumptions and limitations of existing datasets is critical for using them effectively to inform management, enabling both the accurate monitoring of ecosystems and biodiversity, as well as focusing on conservation and management.

## 2. The Post-2020 Framework

The post-2020 global biodiversity framework is the latest round of targets in the field of biological diversity and builds upon a series of previous rounds of targets that had varied levels of success. The most recent iteration of targets preceding the GBF was the Aichi targets; however, none of these were successfully completed based on the original definitions, and one reason for this may have been the inability to effectively monitor progress or designate indicative milestones to ensure progress is “on track” [6].

Within the GBF, there are four goals, which are divided into 23 targets and supported by a series of annexes. The four goals include A—halt habitat loss and prevent extinctions: and has eight targets; B—sustainably use and manage biodiversity: which has four targets; C—digital sequence information: with only a single target; and D—capacity building and resource allocation: with ten targets and ten sub-targets. Most of these targets and all goals are associated with a series of headline indicators and a selection of optional component and complementary indicators.

The headline indicators are meant to be easy to understand and implement in a standardised manner, based on publicly available data, and, in most cases, must be supported by published papers. Most indicators are also aligned with national goals, the SDGs, GEOBONs, or essential biodiversity variables (EBVs). Headline indicators are generally based on national monitoring targets or other national sources, enabling the standardised and scalable tracking of targets at all levels, supported by the optional component and complementary indicators that provide further detail. However, headline indicators do not include all components of each GBF target, and understanding the coverage of data are critical to ensure we have the best data to enable the monitoring of target implementation under the GBF.

## 3. Understanding the Data

The redlists of species and ecosystems are both headline indicators for Goal A of the GBF, and the redlist of species is also the headline indicator for target 4 (preventing the extinction of threatened species). Yet, understanding the limitations of these data are important, as not only does the IUCNs redlist of species represent the main headline indicator for target 4, but it is the main source of data for targets 1–3 for the prioritisation of key areas. However, IUCN and Birdlife data are biased and include errors of omission and commission (overestimating areas whilst missing key areas), as well as strong political biases, especially on the intersection between tropical and subtropical areas [5,7]. There are also issues with inconsistent methodology between regions and taxa, including gridding of range boundaries for reptiles, the use of county boundaries for amphibians in the United States, and the use of river basins in Odonata. This means that the reliable use of data are challenging. Furthermore, the percentage of biodiversity included in spatial plans also needs to be cross-referenced with range maps. The inability to use IUCN and Birdlife data to accurately map species significantly hinders the use of these data, despite being a headline indicator. Furthermore, only a fraction of species are assessed [8], with a significant proportion of species having outdated assessments. Trends for many species may be little more than guess work given the lack of monitoring data, and thus the core species data utilised for species fails to provide the accuracy or detail needed to monitor targets.

The lack of effective data are underscored by assessments conducted within the most recent planet report [1]. Whilst the 69% decrease in populations is likely indicative of significant declines, the lack of consistency in approaches for monitoring both between and within species, as well as the lack of coverage of species and regions, means that understanding genuine population trends is challenging. The absence of long-term data or sites and the lack of any mandate for this within the indicator framework means that these data can only be independently collated within NBSAPS (National Biodiversity Strategic Action Plans). As these data are not mandated or outlined, understanding the dimensions of which data are available is key. 

Most data for the majority of taxa originate from either North America or Europe. For example, 20% of insect data are from the UK, 47% of chordate data are from the United States, and 20% of plantae data are from France (GBIF: https://www.gbif.org/ 25 March 2023). In all cases, these countries lack the majority of species, yet they dominate the distribution of data. Furthermore, almost 90% of distribution data are from within 2 km of a road [4]. In addition, data are largely collected from areas with low elevation and from temperate forests, which are clearly not representative of terrestrial environments. In ocean systems, most data derive from within just kilometres of a coastline, or if not, on a seaway. Therefore, these data cannot be used for mapping diversity, let alone understanding trends or the coverage of biodiversity. This leads to two key questions: Firstly, which data should be used to accurately monitor systems. Secondly, whilst we aggregate data, we need to genuinely understand patterns of diversity and guide protection efforts. What measures can be used to guide interim efforts?

## 4. Bridging the Gaps

As these data are not representative of most regions and taxa, and no adequate data are included in headline indicators of the Monitoring framework, understanding how to use the available data or how to approach research gaps is necessary. Furthermore, monitoring framework notes the essential biodiversity variables (EBVs), which include species distributions and abundances. As noted above, accurate and high-resolution data are needed to meet targets for species conservation. Whilst headline indicators fail to capture high-resolution, species-level data, developing frameworks to guide the collation and aggregation of these data in line with the EBVs would provide the necessary EBVs and better meet targets. Notably, the component and complementary indicators echo elements of species monitoring, but the relevant data simply do not exist at present, and most of the indicators either only cover a subset of species (i.e., Edge Index, which incorporates phylogenetic uniqueness, or assessments for IUCN threatened species). Other indicators within target 4 do not represent actual species-level data, and focus more on agreements. Therefore, understanding how to use the EBVs and incorporating appropriate data into NBSAPs are likely the only ways to monitor level data to bridge the gaps in our understanding of species-level trends.

Firstly, assessments need to account for the level of sampling and focus on the ratio of samples to species [9,10]. Evidently, areas with more data include a greater proportion of present species; however, species accumulation curves are not effective without stipulations to ensure they are representative, including the sample number and degree of coverage [11]. In addition, assessing the degree of completeness is important, and this can be achieved via multiple approaches [2,9]. Another option is to use an index if biodiversity patterns are already being used for prioritisation rather than species-level assessments, provided that index can reconcile the sampling intensity relative to the diversity; however, relatively few diversity assessments have been assessed regarding the minimum data requirements for useful performance. Some of these assessments perform inherently better than others, yet understanding the scenarios under which different metrics perform better is also crucial. Under most scenarios, tested Hill numbers function relatively well, but measuring an average of metrics may be the best approach if indices are being applied, and testing is needed to optimise metric selection [2].

Interpolations provide another approach to assessing patterns of diversity when insufficient data are available for species-level assessments. In interpolations, species-level data are not required; instead, representative site-based assessments are needed to cross-reference with present species using modelling approaches i.e., [11,12]. In these approaches, diversity can be modelled as a baseline for prioritisation or for the targeted assessment of areas with potentially high diversity. These data can also be targeted towards particular landscapes, especially where threat levels may be high, but standardised data on species diversity and their drivers have not been collected in expansive regions [10].

The other option, when data are available, is to develop maps at a species level, though the level of sophistication of species-level analysis depends on the available data. When relatively little data are available, a basic assessment can be conducted by intersecting species needs from point data with a minimum convex polygon of their ranges [9]. These ranges can only be used to make assumptions; they only encompass the edges of sampled range and are only as good as the coverage of available data and inherent biases. Furthermore, any species-level assessment needs to involve clean data, the removal of erroneous records, and the correction of synonyms, provided that good synonym lists are available, as well as spatial thinning if needed and a minimum number of sample points. For higher volumes of data, more sophisticated models can be developed, though efforts to fill spatial and taxonomic data gaps are necessary first [13]. In such models, high-quality representative data need to be paired with appropriate environmental data, and these data require strategic planning to collate and analyse. Based on this analysis, protected area coverage can be calculated on a per-species basis [3]; however, great care is required when handling data since without it, inherent biases mean that outcomes will not be representative across species or regions [4].

## 5. Models and Metrics as Tools for Priority Setting

Goal D of the GBF focuses on capacity and resources, with 19.E focusing on climate biodiversity synergies, and target 8 on climate. Aligning climate and biodiversity targets is crucial, and removing harmful subsidies (T18) may actually be associated with climate targets and the misuse of nature-based solutions, despite new definitions, which have been independently created to remove the existing loopholes that stem from the lack of a glossary in the GBF. Aligning climate and biodiversity targets is possible [14], and, more broadly, other facets of environmental governance can be integrated to reflect the three planetary conditions [15], and to develop frameworks that enable the conservation of biodiversity and ecosystem service provision [16]. However, all of these rely on representative species-level data, which are ultimately needed for the GBF to be usefully fulfilled.

Furthermore, whilst barely referenced within the GBF, the conservation of biodiversity will need to address the risks associated with infrastructural development, which in some regions, poses one of the greatest risks to biodiversity, both causing direct damage and opening up areas to other forms of use [17]. For migratory species in particular, the fragmentation of ranges is a major risk, and collating data to understand species movement patterns is crucial to maintain connectivity in the face of ongoing development [18]. Ensuring that financial mechanisms reflect the ecosystemic risk that may result from new development, as well as developing effective environmental and social impact assessments as requisite components of grants and loans, would provide a mechanism for mediating impacts. Likewise, whilst horizon scanning was removed from some elements of the GBF, developing better precautionary mechanisms for new chemicals and technologies before implementation would prevent many of the issues associated with development, and such precautionary measures are urgently needed. 

## 6. Moving Forwards

The GBF and monitoring framework fail to provide the necessary guidance to fill the gaping data gaps, preventing monitoring diversity from achieving the quality needed to conserve it in the future. However, via the use of EBVs within the monitoring framework, there is an opportunity to develop better frameworks to fill these gaps. Digitising museum collections and providing the means to increase the engagement of citizens both provide the means to fill some of these data gaps. Furthermore, using environmental impact assessments with monitoring as a means to mobilise data in a consistent way would provide a viable means to mobilise data and fill some of these gaps, thus providing a way to assess trends in populations and species ranges. Targeting building capacity and spatial and taxonomic gaps is also necessary. Although these factors fall outside of the scope of defined indicators, using the working groups of GEOBON could help mediate this.

The GBF represents a successor to the largely unsuccessful Aichi targets. Whilst the Aichi targets undoubtedly facilitated some progress towards various conservation goals, especially around protected area coverage, they failed to be completed, in part because of a lack of guiding metrics to assess progress. To reconcile this, the monitoring framework was meant to bridge this gap and provide key metrics. However, given the gaps between the required data, existing data, and targets, additional approaches to aggregate better data will be needed at all levels. Ultimately, given that the GBF is meant to provide a foundation and baseline for new targets in 2030, we need to lay the groundwork, especially for data infrastructure, paving the way for a better future and evidence-based targets for global biodiversity.

## Data Availability

The data is contained within the manuscript.

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
