# Peer review of "Developing Biodiversity Baselines to Develop and Implement Future Conservation Targets"

_plants, 2023, doi:10.3390/plants12122291_

Round 1
Reviewer 1 Report
Dear Author,
I have read Your manuscript with great pleasure. In my opinion it is generally well-written, however I have found suggestions for its improvement. In my opinion ich chapter Introduction might be added brief overview of causes of diminishing global biodiversity (i.e biological invasions, global warming, loss of semi-natural habitats due to lack of traditional management etc.). The way of incorporation of literature sources should be corrected according to Instructions for Authors.
Author Response
1
I have read Your manuscript with great pleasure. In my opinion it is generally well-written, however I have found suggestions for its improvement. In my opinion ich chapter Introduction might be added brief overview of causes of diminishing global biodiversity (i.e biological invasions, global warming, loss of semi-natural habitats due to lack of traditional management etc.). The way of incorporation of literature sources should be corrected according to Instructions for Authors.
Response: Thank you, references and drivers of biodiversity loss have been added as suggested
Reviewer 2 Report
The aim of the paper is to highlight the inability of currently available data to fulfil the goals set by the Post-2020 Global Biodiversity Framework, and how the targets within the framework itself might be improved and implemented. The article mentions a series of major criticalities and gaps in species data, and the uneven distribution of valuable data among different countries and social contexts. Lastly, the paper gives proper suggestions on how to resolve these issues in order to achieve more effective conservation practices.
The title is simple and catchy, but it might need some revision, since it does not accurately reflect the content of the article itself: the paper focuses more on an assessment of the current situation, and the importance of the development of the baselines mentioned in the title itself.
In fact, while some valuable suggestions are given on different aspects of the development, some terms are used in the paper without contextualisation, e.g. modelling approaches and indexes are mentioned, but it is not clear what these models and indexes are, how they can be applied within the framework, and thus, how inherent they are to the topic.
It is advised that these aspects are further investigated for the aim of the paper, in order to give a more in-depth idea of how these can be applied for taxa and geographic regions more accurately.
The references are mostly appropriate; however, some claims might need to be supported by proper source, e.g.: “these analyses of trends result from aggregations of inconsistent data which is not truly representative across regions or taxa” (page 1). Furthermore, while some major references are mentioned, it is advised that the bibliography is deepened in order to give a broaden view of the topic.
Another issue is the abbreviation of Essential Biodiversity Variables, mostly referred to as “EGV” instead of “EBV”.
As concerns some minor issues with the writing, the terminology is mostly accurate and is indicative of a proper knowledge in the research area; the writing flow, however, needs some improvement. In fact, the purpose of some sentences results unclear, and some paragraphs might need a revision and rearrangement of their structure, e.g. “Ensuring financial mechanisms reflect ecosystemic risk that may result from new development and developing effective environmental and social impact assessments as requisite components of grants and loans would provide a mechanism for mediating impacts” (page 4).
This aspect is also exacerbated by the improper use of punctuation, which makes some sentences hard to read and contextualise, e.g. “With the recent launch of the post-2020 framework, and associated monitoring framework understand the framework, and the data needed to support it is crucial” (page 1).
Lastly, some grammar mistakes are present throughout the paper, e.g. “to have publically available” (page 2), “the percent of biodiversity” (page 2), “it is includes errors” (page 3).
For these issues, a thorough revision by a professional editor is recommended before publication.
Author Response
The title is simple and catchy, but it might need some revision, since it does not accurately reflect the content of the article itself: the paper focuses more on an assessment of the current situation, and the importance of the development of the baselines mentioned in the title itself.
In fact, while some valuable suggestions are given on different aspects of the development, some terms are used in the paper without contextualisation, e.g. modelling approaches and indexes are mentioned, but it is not clear what these models and indexes are, how they can be applied within the framework, and thus, how inherent they are to the topic.
It is advised that these aspects are further investigated for the aim of the paper, in order to give a more in-depth idea of how these can be applied for taxa and geographic regions more accurately.
Response: Thank you. More than half the paper focuses on metrics for monitoring and developing baselines, so the title does reflect what the article aims to do, though obviously assessing the current knowledge state is crucial to actually move forwards. However some additional lines have been added for clarity where needed
The references are mostly appropriate; however, some claims might need to be supported by proper source, e.g.: “these analyses of trends result from aggregations of inconsistent data which is not truly representative across regions or taxa” (page 1). Furthermore, while some major references are mentioned, it is advised that the bibliography is deepened in order to give a broaden view of the topic.
Response: References have been added to support the statement. The paper aims to provide a succinct overview of data needs and gaps for the Kunming-Montreal Global Biodiversity framework, so much of the paper is explaining the framework and therefore does not require a huge number of references to support such discussion.
Another issue is the abbreviation of Essential Biodiversity Variables, mostly referred to as “EGV” instead of “EBV”.
Response: Thank you, this has been corrected
Comments on the Quality of English Language
As concerns some minor issues with the writing, the terminology is mostly accurate and is indicative of a proper knowledge in the research area; the writing flow, however, needs some improvement. In fact, the purpose of some sentences results unclear, and some paragraphs might need a revision and rearrangement of their structure, e.g. “Ensuring financial mechanisms reflect ecosystemic risk that may result from new development and developing effective environmental and social impact assessments as requisite components of grants and loans would provide a mechanism for mediating impacts” (page 4).
This aspect is also exacerbated by the improper use of punctuation, which makes some sentences hard to read and contextualise, e.g. “With the recent launch of the post-2020 framework, and associated monitoring framework understand the framework, and the data needed to support it is crucial” (page 1).
Response: The Author is English. I apologise for the typos, or somewhat arcane sentence construction in some elements, however these are still grammatically correct. However, the document has been proofed to ensure that sentences are comprehensible throughout
Lastly, some grammar mistakes are present throughout the paper, e.g. “to have publically available” (page 2), “the percent of biodiversity” (page 2), “it is includes errors” (page 3).
Response: Thank you, it has now been proofed to remove such errors and typos
For these issues, a thorough revision by a professional editor is recommended before publication.
Response: minor typos have been fixed, and commas added as appropriate